# The Effect of Er:YAG Lasers on the Reduction of Aerosol Formation for Dental Workers

**DOI:** 10.3390/ma14112857

**Published:** 2021-05-26

**Authors:** Kinga Grzech-Leśniak, Jacek Matys

**Affiliations:** 1Laser Laboratory, Department of Oral Surgery, Wroclaw Medical University, 50-425 Wroclaw, Poland; jacek.matys@wp.pl; 2Department of Periodontics, School of Dentistry, Virginia Commonwealth University, Richmond, VA 23298, USA

**Keywords:** COVID-19, dentistry, erbium laser, Er:YAG, high-volume evacuator, SARS-CoV-2

## Abstract

Infection prevention in dental practice plays a major role, especially during the COVID-19 pandemic. This study aimed to measure the quantity of aerosol released during various dental procedures (caries and prosthetic treatment, debonding of orthodontic brackets, root canal irrigation) while employing the Er:YAG lasers combined with a high-volume evacuator, HVE or salivary ejector, SE. The mandibular second premolar was extracted due to standard orthodontic therapy and placed in a dental manikin, to simulate typical treatment conditions. The particle counter was used to measure the aerosol particles (0.3–10.0 μm) at three different sites: dental manikin and operator’s and assistant’s mouth area. The study results showed that caries’ treatment and dental crown removal with a high-speed handpiece and the use of the SE generated the highest aerosol quantity at each measured site. All three tested Er:YAG lasers significantly reduced the number of aerosol particles during caries’ treatment and ceramic crown debonding compared the conventional handpieces, *p* < 0.05. Furthermore, the Er:YAG lasers generated less aerosol during orthodontic bracket debonding and root canal irrigation in contrast to the initial aerosol quantity measured in the dental office. The use of the Er:YAG lasers during dental treatments significantly generates less aerosol in the dental office setting, which reduces the risk of transmission of viruses or bacteria.

## 1. Introduction

Dental workers are exposed to the risk of viral airborne infection [1,2]. With the COVID-19 wave running through the world, all medical professionals are at the highest risk of infection by the SARS-CoV-2, which causes sickness ranging from symptoms like the common cold to severe respiratory diseases because of the direct contact of the virus with the eyes, nose, and oral cavity through their mucous membranes as the main infection route [3]. A common feature of dental professional activity is working in the environment where human bioaerosols mix with water sprays, the particles of which then increase their velocity and scatter in the office [4,5,6,7]. The primary sources of contamination in dental offices are respiratory particles (breathing, coughing, sneezing) and saliva mixed with patients’ blood. The second source that can spread the infection in the office are rotary instruments such as high- and low-speed handpieces, dental sandblaster, ultrasonic scalers, or lasers [3,8].

Most dental procedures using handpieces generate aerosols [9]. The highest aerosol concentration is observed during treatment performed by a contra-angle handpiece [3,10,11]. Among them, dental procedures with the highest splatter and aerosol generation are associated with conservative treatment, tooth prosthetic preparation, and ultrasonic scaling [12]. The aerosols produced during dental procedures can be divided into three subgroups [6]. The first group contains respiratory aerosols formed while sneezing or coughing. The water sprays produced by dental rotary instruments, ultrasonic tools, and lasers constitute the second group. The last group contains a mixture of water sprays with respiratory aerosols that have a high velocity and can be spread quickly in the office. The aerosols contain a mixture of air with water particles greater than 50 μm formed during splatter [11]. The most prominent hazard in the dental office is smaller particles with diameters of 0.5 to 10 μm. These particles can be transported when they are inhaled to terminal bronchioles and alveoli of the human lungs [4].

Healthcare professionals have the responsibility to introduce updated innovations with safety protocols for the prevention of transmission of all infections to patients and staff. Additionally, it is more crucial today in this current worldwide COVID-19 pandemic. Our consideration was to focus on lasers that we can apply in many dental treatment procedures [13,14,15]. Among different wavelengths used in dentistry, the highest efficiency in treating both hard and soft tissues have erbium family lasers [16,17]. Er:YAG (2940 nm) and Er,Cr:YSGG (2780 nm) lasers have excellent absorption coefficient in water and, thus, are useful in treating tooth caries, ablating bone, and vaporizing soft tissues of the oral mouth [18,19,20,21]. The laser irrigation increases water particles vibration, which raises the thermal energy of the water and causes its evaporation after exceeding the temperature of 100 °C [20,22]. The photothermal effect produced by high-power lasers allows efficient cutting of hard and soft tissues [23]. However, erbium laser operation in a water spray, which is needed for tissue cooling, carries a potential risk of spreading the virus. Moreover, in the medical market we can find a different erbium laser with various cooling systems that can raise aerosols’ concentration. Thus, the assessment of the number of aerosols released during dental treatments when using various erbium laser systems should be addressed.

The amount of aerosol produced by conventional dental handpieces is related to the rotor torque speed and the water pressure delivered to the tip [24]. Although the operation with erbium lasers requires continuous cooling with a water spray, the lack of rotation and vibration of the laser tip itself, placed in the applicator socket, does not cause additional centrifugal dispersion of water spray particles. The principal operation of erbium lasers in tissues is based on similar effects. Notwithstanding that, various cooling supply methods used in erbium lasers available on the medical market may result in different aerosol particle sizes and amounts released. The cooling system installed in laser devices differs in delivering the coolant to the working tip, the type of coolant transported, the length of the liquid supply lines, and the type of compressors used. Different erbium family lasers have been provided within one or more supply lines having exit canals in the handpiece for compressed air and water’s concurrent application to the target tissue. This feature regulates the flow of the sprays [25,26].

This work aimed to test a null hypothesis that there is no difference in the number of aerosols generated by three various Er:YAG lasers compared to conventional high- and low-speed handpiece for tooth preparation. Furthermore, the differences in aerosol generation among the lasers during prosthetic crowns’ retrieval, orthodontic ceramic brackets debonding, and endodontic root canal irrigation was tested.

## 2. Materials and Methods

The experiment was conducted using a dental manikin head. In the area of a lower-left second premolar (35 FDI classification, World Dental Federation), a natural tooth extracted due to standard orthodontic therapy was placed to simulate typical treatment conditions. The measurement of generated aerosols was done at three sites: (1) 2 cm from manikin’s mouth, (2) 2 cm from the assistant’s mouth, and (3) 2 cm from the operator’s mouth. The distance from the manikin’s mouth to the operator and the assistant’s mount was measured with a ruler and amounted to 40 cm (Figure 1). The protocol of the experiment was prepared by the authors of the paper.

### 2.1. Aerosol Measurement Protocol

The PC200 laser counter was applied (Trotec GmbH, Schwerin, Germany) to estimate the aerosol particles at the conducted sites. The particle sensor works by measuring the light dispersion effect in a medium. The intensity of scattered light is measured at a specified angle in particulate matter, allowing calculation of the aerosol particles’ amount. The particle counter used in the test had the current calibration certificate issued by the manufacturer and was compliant with the ISO 21501-4 standard, which specifies the measurement accuracy of 95%. The PM200 sensor allowed us to measure six aerosol fractions with a diameter of 0.3–10.0 µm (0.3, 0.5, 1.0, 2.5, 5.0, 10.0 µm). The nozzle of the counter was placed 2 cm from the operator’s, assistant’s, and manikin’s mouth. The PM 200 sensor was turned 1 min after the beginning of each procedure. Time measurements were done using a stopwatch. Each measurement was repeated six times during the experiment. The number of measured particles (different fractions) was summarized, and the mean results found in six repeated measurements were compared among study groups.

### 2.2. Technical Features of Er:YAG Lasers and Cooling System Applied in These Devices

The characteristic of three common erbium lasers used in dentistry were compared in the study, as shown in Table 1. The lasers applied in the study were equipped with a cooling delivery system, described in Figure 2. One of the lasers applied in the study (AdvErL Evo) used 0.7% saline solution as a coolant. The saline solution was transported from the container located inside the device by a supply line to the exit located in the modified laser tip. The target tissue was cooled by only a single coolant stream. The other two types of erbium lasers (LiteTouch and LightWalker lasers) were used to transport the coolant (distilled water) supply lines, with three ends located in the handpiece’s head. Thus, when using these two lasers the tissue was cooled by three streams of water.

### 2.3. Dental Treatment Procedures and Suction Systems Used in the Study

All procedures were repeated six times. The quantity of dental aerosol particles was measured during the following procedures.

#### 2.3.1. Caries’ Removal

Caries’ class I treatment by means of (1) the round diamond bur (#014) with a high-speed handpiece W&H Synea TA-98LC (W&H, Bürmoos, Austria) at working parameters of 200,000 RPM (revolutions per minute) and water cooling of 30 mL/min; (2) the round rose bur (#014) with a low-speed handpiece W&H Synea TA-98LC (W&H, Bürmoos, Austria) at working parameters of 15,000 RPM and water cooling of 30 mL/min; (3) the contra-angle handpiece of Er:YAG laser (AdvErL Evo, Morita, Kyoto, Japan) with laser parameters of energy 300 mJ, frequency 10 Hz, power 3 W, tip diameter 1 mm, and water/air coolant 10/7; (4) the contra-angle handpiece H14 of Er:YAG laser (LightWalker, Fotona, Ljubljana, Slovenia) with laser parameters of energy 300 mJ, frequency 20 Hz, power 6 W, MSP mode, tip diameter 1 mm, and water/air coolant 6/4; and (5) the contra-angle handpiece of Er:YAG laser (LiteTouch, LightInstruments, Yokneam, Israel) with laser parameters of energy 300 mJ, frequency 20 Hz, power 6 W, tip diameter 1 mm, and water/air coolant 8 (Figure 3).

#### 2.3.2. Crown Laser Debonding

The simulation of cutting the crown was performed by applying a diamond round bur on a high-speed contra-angle to the buccal surface of the tooth (control) for 1 min at working parameters: of 15,000 RPM and water cooling of 30 mL/min. The tooth crown was irradiated with tested lasers using a non-contact, continuous mode occlusally, buccally, lingually, and indirectly interproximally at a distance of 5–10 mm. Following 1 minute after each removal/debonding procedure, the aerosols’ measurement was started with the PM200 counter. The parameters used for prosthetic debonding by different lasers were as follows.
(1)The contra-angle handpiece H14 of Er:YAG laser (LightWalker, Fotona, Slovenia) with laser parameters of energy 300 mJ, frequency 15 Hz, power 4.5 W, MSP mode, tip diameter of 1 mm, and water/air coolant 4/4;(2)The contra-angle handpiece of Er:YAG laser (AdvErL EVO, Morita, Japan) with laser parameters of energy 350 mJ, frequency 10 Hz, power 3.5 W, tip diameter: of 1 mm, and water/air coolant 4/4;(3)The contra-angle handpiece of Er:YAG laser (LiteTouch, LightInstruments, Israel) with laser parameters of energy 300 mJ, frequency 15 Hz, power 4.5 W, tip diameter of 1 mm, and water/air coolant 8/8 (Figure 4).


#### 2.3.3. Orthodontic Brackets’ Debonding

A metal bracket (Victory Series, 3M Unitek, Monrovia, CA, USA) was bonded, according to the manufacturer’s recommended protocols, on the second premolar tooth’s labial surfaces using orthodontic composite adhesive Transbond XT (3M Unitek, Monrovia, CA, USA). The bracket debonding procedure was done according to the previously published paper [22]. That procedure was repeated six times and the amount of the aerosol was measured using the PM200 counter. The lasers’ parameters used for orthodontic brackets’ debonding were as follows.(1)The contra-angle handpiece H14 of Er:YAG laser (LightWalker, Fotona, Slovenia) with laser parameters of energy 170 mJ, frequency 20 Hz, power 3.4 W, MSP mode, tip diameter of 1 mm, and water/air coolant 3/3;(2)The contra-angle handpiece of Er:YAG laser (AdvErL EVO, Morita, Japan) with laser parameters of energy 170 mJ, frequency 20 Hz, power 3.4 W, tip diameter of 1 mm, and water/air coolant 3/3;(3)The contra-angle handpiece of Er:YAG laser (LiteTouch, LightInstruments, Israel) with laser parameters of energy 150 mJ, frequency 25 Hz, power 3.75 W, tip diameter of 1 mm, and water/air coolant 2/2 (Figure 5).


#### 2.3.4. Endodontic Treatment

The second premolar tooth’s chamber was opened with a dental round bur on a high-speed dental handpiece, and the root canal was prepared with ISO 30/06 file. The root canal irrigation was carried out using endodontic needle EndoEze (Ultradent, South Jordan, UT, USA) placed to the root canal 1 mm before its working length. The laser irrigation was done by inserting a laser tip and the endodontic needle in the middle part of the chamber, according to the photon-initiated photoacoustic streaming protocol. [27] The 2% sodium subchloride solution was pushed into the root canals by the laser cavitation. The laser parameters used for endodontic irrigation were as follows.(1)The contra-angle handpiece H14 of Er:YAG laser (LightWalker, Fotona, Slovenia) with laser parameters of energy 10 mJ, frequency 15 Hz, power 0.15 W, SSP mode, tip diameter of 0.6 mm, and water/air coolant 0/0;(2)The contra-angle handpiece of Er:YAG laser (AdvErL EVO, Morita, Japan) with laser parameters of energy 30 mJ, frequency 10 Hz, power 0.3 W, tip diameter of 0.6 mm, and water/air coolant 0/0;(3)The contra-angle handpiece of Er:YAG laser (LiteTouch, LightInstruments, Israel) with laser parameters of energy 40 mJ, frequency 10 Hz, power 0.4 W, tip diameter of 0.6 mm, and water/air coolant 0/0 (Figure 6).


### 2.4. Spray/Aerosol Evacuators

The following suction systems were used to remove the aerosols produced when applying different dental procedures: (1) saliva ejector (SE) EM15 (Monoart^®^ Euronda, Vicenza, Italy) and (2) high-volume evacuator (HVE) EM19 EVO (Monoart^®^ Euronda, Vicenza, Italy). Evacuators were placed at the level of the tooth around 2 cm from its buccal side.

### 2.5. The Office Air Standardization

All measurements were made in a closed room (dentist’s office) with an area of 20 m^2^. During the research, all doors and windows were closed and the air conditioning was turned off. Before each test, the number of particles in the room was tested, and the next measurements were made if particles’ numbers were in the range of 28,000–30,000. The stable value of particles in the room was maintained with the air purifier system (NV1050, Novaerus, Dublin, Ireland) with an air exchange of 800 m^3^ per hour. Control measurements were taken every 1 min while the purifier was on. After the assumed number of particles in the room was obtained, the given procedure (measurement) was performed. The control measurement of particles in the room was made after placing the sensor in the office’s center. The average time to purify the air to the demanded particle range in the office (20 m^2^) was around 5 min.

### 2.6. Statistical Analysis

The total number of aerosol particles in a range of 0.3–10 um formed during different dental procedures by various therapeutic methods was measured in three areas and compared with the analysis of variance ANOVA (Analysis of Variance) analysis with post hoc tests (multiple comparisons using the Tukey test). Statistica software (StatSoft, Tulsa, OK, USA) was used for statistical analysis. Values below *p* = 0.05 were considered to be statistically significant.

## 3. Results

### 3.1. Erbium Lasers Reduced the Number of Aerosol Particles during Caries’ Treatment

The aerosol levels measured at the manikin’s, assistant’s, and operator’s mouths were significantly lower for tested laser systems when compared with conventional contra-angle handpieces during caries ‘removals, *p* < 0.001. Caries’ removal by using a high- and low-speed handpiece combined with HVE resulted in significant aerosols’ decrease in contrast to the SE, *p* < 0.001 (Table 2). Interestingly, we found a similar particle number for all the lasers combined with the saliva ejector or high-volume evacuator except the LightTouch laser. The particle number was significantly higher when the LightTouch laser with HVE was applied at the operator’s mouth in contrast to the assistant’s and manikin’s mouth levels, *p* < 0.05 (Figure 7).

### 3.2. Erbium Lasers Reduces the Number of Aerosol Particles during Ceramic Crown Debonding

The number of aerosols generated during prosthetic ceramic crown debonding was characterized by its significant reduction using erbium lasers in contrast to conventional crown removal by the dental turbine, *p* < 0.001. Comparison of the erbium lasers combined with SE and HVE showed a significant decrease in aerosols’ generation for the Morita laser at operator and assistant levels, *p* < 0.001. However, application of all the lasers with HVE for crown debonding resulted in insignificant differences in aerosols’ generation at the manikin’s mouth, *p* > 0.05 (Table 3).

### 3.3. Erbium Lasers Generated a Minimal Quantity of Aerosols during Orthodontic Bracket Debonding

The highest particle number, amounting to 31.4 (SD1.5), was found for the LiteTouch laser. The lowest level of aerosol particles was found for the Morita laser, as compared with LightWalker and LiteTouch lasers at the operator and assistant levels for both suction systems, *p* < 0.001. The results of aerosols at the manikin mouth level were similar, with no significant differences among lasers, *p* > 0.05 (Table 3). Comparison of suction tools efficiency used during the debonding procedure showed a significant decrease in the aerosol amount for HVE at the manikin and operator levels (LightWalker laser) and the manikin’s mouth for LightTouch laser, *p* < 0.05 (Table 4).

### 3.4. Erbium Lasers-Assisted Endodontic Irrigation Generating Minimal Amount of Aerosols

Evaluation of aerosols during a root canal irrigation with all tested laser systems showed insignificant differences when compared to the endodontic needle irrigation alone. Furthermore, we found no differences in aerosol quantity measured at different sites, using different suction devices during root canal irrigation, *p* > 0.05 (Table 5).

## 4. Discussion

Reducing the aerosol quantity while working in the dental practice is essential to diminish the risk of SARS-CoV-2 virus transmission during the COVID-19 pandemic. Working with a high-speed dental turbine generates many aerosol particles; therefore, it is crucial to use the aerosol removal systems simultaneously. Another method for decreasing aerosolization during dental treatment is to replace classical dental handpieces with hard-tissue lasers. The study results showed that caries’ removal with high- and low-speed handpieces and saliva ejector generated the highest amount of spray particles at each measured site. The use of high-volume evacuators significantly reduced aerosols during caries’ treatment in contrast to the salivary ejector. Application of Er:YAG lasers decreased aerosol generation around two times compared to both conventional handpieces at the dental manikin’s mouth. All tested lasers generated similar low aerosols during prosthetic crown and orthodontic brackets’ debonding and root canal irrigation. These findings are of importance since they point toward the efficiency and safety of using Er:YAG laser in dental treatment, especially during a pandemic.

The study’s main aim was to test a null hypothesis that there is no difference in aerosol quantity generated during dental caries’ treatment using conventional rotary instruments and various erbium lasers. The study rejected the null hypothesis. All the lasers in the experiment significantly decreased the aerosols produced during caries’ removal. The mean results of aerosol release for all the lasers were in the range of 29,400 to 35,000 particles; thus, around 7, 5, 2, and 1.5 times less than high- and low-speed handpieces combined with SE or HVE, respectively. In the current literature, there is no research evaluating the production of aerosol during dental procedures using Er:YAG lasers. However, the effect of different suction systems was clearly described for conventional handpieces [4,12,28]. The study of Jacks [12] showed that the HVE eliminated approximately 90% of aerosols during ultrasonic scaling in contrast to conventional SE. Moreover, Rupf et al. [28] recommended HVE to reduce patients’ and dental staffs’ exposure to fine and ultrafine airborne particles when using scanning sprays. Furthermore, Naulty et al. [29] and Matys and Grzech-Leśniak [30], in their studies, recorded aerosol particulate at statistically significantly increased levels during dental procedures. Our study also found an average 2- to 3-fold reduction in aerosol quantity when using HVE compared to SE, which confirms the importance of the proper suction system during tooth preparation (treatment of Black’s class I caries) in order to increase microbial safety in the dental office.

A dental treatment, particularly while using high-speed handpieces, produces a high quantity of aerosols and splatters, possibly contaminated with bacteria, viruses, fungi, and blood [31]. The aerosol generated in the dental office during treatment consists of particles smaller than 50 µm. Particles below 10 µm pose a particular risk of acquiring viral infection and transmitting the virus by inhalation among patients and dental office staff [9,11,32]. The increase of aerosol quantity in the dental office is related to the length of the treatment procedure and the quality of the suction system. One of the dental procedures with a greater risk of a high aerosol formation is prosthetic preparation. Ceramic crowns’ debonding procedure can be also accomplished using erbium family lasers [33,34,35]. In the present study, we compared the aerosol particle sizes when cutting a ceramic prosthetic crown with a bur on a dental turbine and crown debonding with the Er:YAG lasers. We found a significantly lower aerosol quantity generated during crown removal using all Er:YAG lasers in contrast to the high-speed turbine with both HVE and SE. There are no other studies in the literature confirming our results for the Er:YAG lasers used in dental treatment. However, studies by Harrel et al. [6] and Jacks [12] support our results, showing better efficiency of HVE compared to SE for aerosols’ reduction.

The operation of erbium lasers in tissues is based on the absorption of laser energy by water, which leads to an increase in the vibration of hydrogen molecules and an increase in the thermal energy [19,36]. This process causes the water to evaporate and induces movements within the liquid after the laser tip is placed in it. The cavitation effect caused by the photoacoustic phenomenon was used to clean the debris in root canals during endodontic treatment. However, this effect could also potentially create aerosols during dental treatment. Our study results indicated that the laser action induced in the tooth chamber filled with sodium hypochlorite administered with a syringe had a negligible difference in aerosol compared to irrigation with an endodontic needle alone. It should be emphasized that endodontic treatment performed according to FDI indications (using a cofferdam, rinsing the mouth with antiseptic fluids) and when creating access to the chamber using erbium lasers can be one of the safest procedures in dentistry carried out during the COVID-19 pandemic [10,37,38].

In the conducted study, we used three lasers with the same electromagnetic wavelength (2940 nm). The principal operation of these lasers in tissues is based on similar physical phenomena. However, differences in the methods of supplying cooling fluid to the tissue may result in different aerosol particle formation during dental treatment. Our study showed less aerosol increase when the AdvErL EVO (Morita, Kyoto, Japan) laser was used compared to LightWalker (Fotona, Ljubljana, Slovenia, *p* > 0.05) and LightTouch (Light Instruments, Yokneam, Israel *p* < 0.05) lasers during the crowns’ debonding procedure. The main difference between the lasers was that the water and air supply of the AdvErL EVO laser were combined into the tip (not only in the handpiece), where the laser rays were transmitted and emitted, and also in individual supply lines [25]. In addition, coolant from the AdvErL Evo laser to the tissue was transported through only one discharge channel, which created a continuous stream of water. In the other two lasers, the water–air spray was led through three channels to the tip, which may result in the formation of more aerosols. However, it should be emphasized also that the maximum power of the Morita laser was 4 W. Thus, in treatments where higher power to decrease operation time is needed (greater thickness of crowns, removal of bridges), the procedure itself may take longer and, consequently, more aerosols may be formed. The laser light interaction with the target tissue has some additional benefits in reducing infection risk in dentistry. The main limitation of using erbium lasers versus conventional rotary instruments in dentistry is their high price. Moreover, working with these devices also requires gaining experience and knowledge for safe work in vivo. Additional studies should be conducted to investigate decontamination efficiency of Er:YAG laser during dental treatment. Furthermore, randomized clinical trials concerning the effect of lasers on aerosols’ reduction are needed.

## 5. Conclusions

The use of erbium lasers during dental treatments (cavity preparation, full ceramic crown debonding, orthodontic brackets’ debonding) significantly reduces the aerosol amount in the dental office and should be used alternatively to conventional rotatory tools, especially during the COVID-19 pandemic.

## Figures and Tables

**Figure 1 materials-14-02857-f001:**
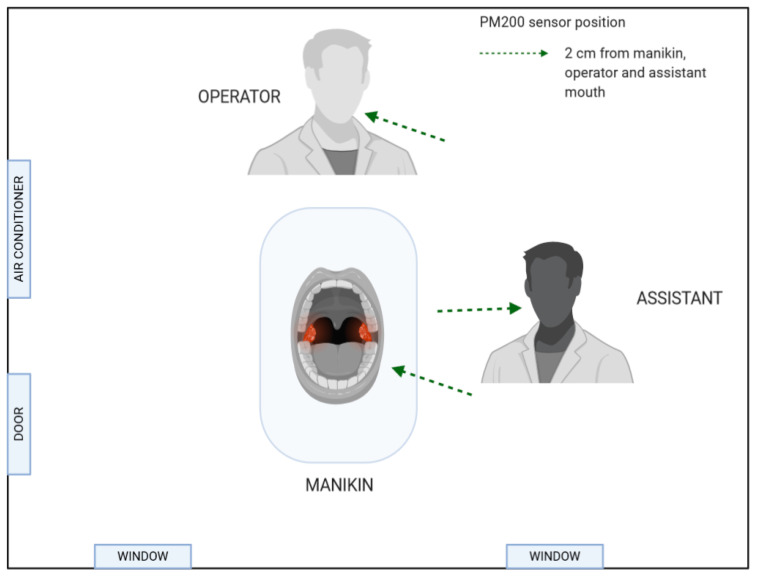
Aerosol particle sensor positions (created in BioRender.com).

**Figure 2 materials-14-02857-f002:**
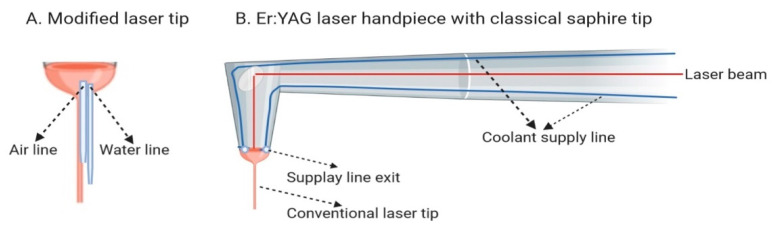
Types of coolant supply in erbium family lasers. (**A**) Modified laser tip with water and air supply lines built in the tip. (AdvErL Evo laser); (**B**) Conventional laser handpiece with water supply lines in the handpiece and with the lines’ exits in the head of the handpiece (next to the tip socket (LiteTouch and LightWalker lasers).

**Figure 3 materials-14-02857-f003:**
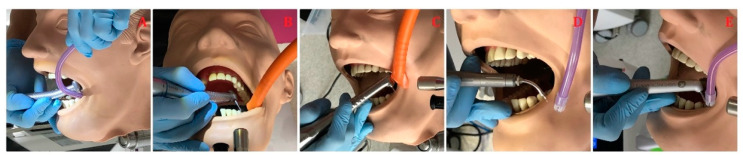
Caries’ removal by conventional handpieces and Er:YAG lasers: (**A**) a high-speed handpiece; (**B**) a low-speed handpiece; (**C**) LightWalker; (**D**) AdvErL Evo; (**E**) LiteTouch.

**Figure 4 materials-14-02857-f004:**
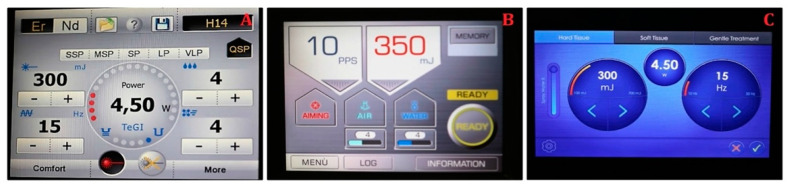
Crown debonding with Er:YAG lasers: (**A**) LightWalker (**B**) AvErL Evo (**C**) Litetouch.

**Figure 5 materials-14-02857-f005:**
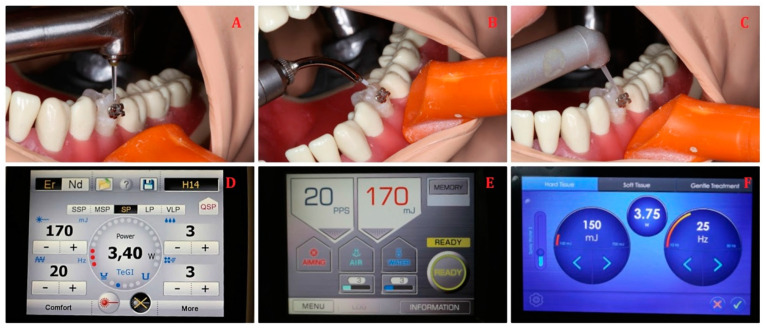
Orthodontic brackets debonding with Er:YAG lasers: (**A**,**D**) LightWalker; (**B**,**E**) AdvErL Evo; (**C**,**F**) LiteTouch.

**Figure 6 materials-14-02857-f006:**
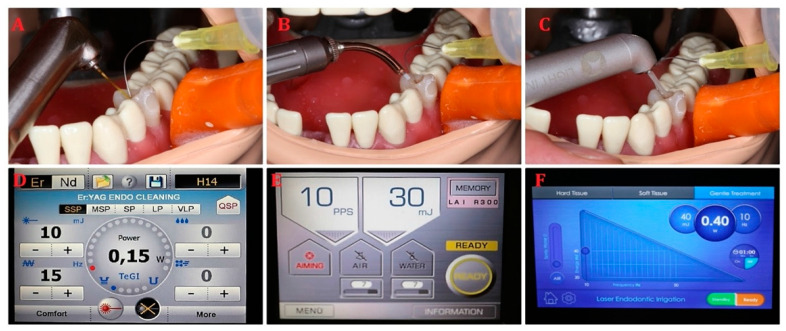
Root canal irrigation with Er:YAG lasers: (**A**,**D**) LightWalker; (**B**,**E**) AdvErL Evo; (**C**,**F**) LiteTouch.

**Figure 7 materials-14-02857-f007:**
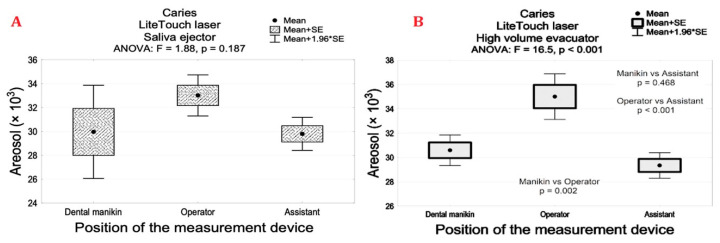
The level of aerosol particles (×103) measured at the manikin’s, operator’s, and assistant’s mouths for the LightTouch Er:YAG laser with (**A**) salivary ejector and (**B**) high-volume evacuator. (Mean (SD)) during tooth/caries’ preparation.

**Table 1 materials-14-02857-t001:** The characteristic of three common erbium lasers used in dentistry and compared in the study.

–	AdvErL Evo Laser	LiteTouch Laser	LightWalker Laser
Laser class	class 4	class 4	class 4
Mode	pulsed	pulsed	pulsed
Wavelength	2940 nm	2940 nm	2940 nm
Transmission system	flexible waveguide	reduced articulated arm	articulated arm
Air compression system	built in	external	built in
Types of coolant supply	modified laser tip with water and air supply lines built in the tip	conventional laser handpiece with water supply lines in the handpiece and with the lines’ exits in the head of the handpiece	conventional laser handpiece with water supply lines in the handpiece and with the lines’ exits in the head of the handpiece
Coolant	saline solution	distillated water	distillated water

**Table 2 materials-14-02857-t002:** The level of aerosol particles (×103) measured at the manikin’s, operator’s, and assistant’s mouths (Mean (SD)) during tooth/caries’ preparation.

Procedure	Tools	Exhaustion	Place of Measurement			ANOVA *p*
			Manikin (A) Mean (SD)	Operator (B) Mean (SD)	Assistant (C) Mean (SD)	*p* Values
Caries	High-speed handpiece	Saliva ejector	235.2 (18.80)	112.2 (18.6)	101.5 (10.6)	A vs. BC *p* < 0.001
		High volume evacuator	64.1 (4.6)	42.3 (4.5)	33.5 (3.1)	A vs. BC *p* < 0.001 B vs. C *p* < 0.001
	Low-speed handpiece	Saliva ejector	183.0 (8.1)	89.1 (7.6)	44.3 (4.9)	A vs. BC *p* < 0.001 B vs. C *p* < 0.001
		High volume evacuator	55.1 (3.3)	37.1 (4.2)	34.1 (4.5)	A vs. BC *p* < 0.001
	Morita laser	Saliva ejector	30.1 (0.7)	30.0 (0.8)	29.6 (0.5)	0.447
		High volume evacuator	29.5 (0.6)	29.7 (0.9)	29.2 (0.4)	0.542
	Fotona laser	Saliva ejector	29.9 (1.3)	32.5 (3.8)	32.1 (1.5)	0.368
		High volume evacuator	29.4 (1.4)	33.8 (2.8)	31.3 (1.5)	0.248
	LiteTouch laser	Saliva ejector	29.9 (4.9)	33.0 (2.1)	29.8 (1.7)	0.187
		High volume evacuator	30.6 (1.6)	35.0 (2.3)	29.3 (1.3)	B vs. AC *p* < 0.05
–	–	ANOVA	*p* < 0.001	*p* < 0.001	*p* < 0.001	—

**Table 3 materials-14-02857-t003:** The level of aerosol particles (×103) measured at manikin, operator, and assistant mouths (Mean (SD)) during ceramic crown debonding.

Exhaustion	Place of Measurement	Crown Debonding—Tools				ANOVA *p*
		Morita Laser (A) Mean (SD)	Fotona Laser (B) Mean (SD)	LiteTouch Laser (C) Mean (SD)	High-Speed Turbine (D)	*p* Values
Saliva ejector	Manikin	41.8 (2.7)	43.6 (2.2)	54.3 (4.7)	284.8 (26.5)	D vs. ABC *p* < 0.001 C vs. AB *p* < 0.001
	Operator	33.1 (2.7)	44.1 (2.6)	47.4 (3.9)	154.4 (22.3)	D vs. ABC *p* < 0.001 B, C vs. A *p* < 0.001
	Assistant	30.7 (1.4)	43.2 (2.1)	43.7 (3.1)	112.5 (14.2)	D vs. ABC *p* < 0.001 B, C vs. A *p* < 0.001
High volume evacuator	Manikin	40.5 (1.9)	43.2 (0.6)	44.1 (4.0)	67.5 (8.3)	D vs. ABC *p* < 0.001
	Operator	30.6 (0.6)	43.9 (1.8)	44.3 (3.1)	58.3 (7.4)	D vs. ABC *p* < 0.001 B, C vs. A *p* < 0.001
	Assistant	29.4 (1.2)	43.6 (2.4)	43.1 (2.9)	47.5 (4.3)	D vs. ABC *p* < 0.001 B, C vs. A *p* < 0.001

**Table 4 materials-14-02857-t004:** The level of aerosol particles (×103) measured at manikin, operator, and assistant mouths (Mean (SD)) during orthodontic bracket debonding. Similar small letters in a column indicate statistical significance between suction tools assessed at the same measured level (operator, assistant, or manikin).

Exhaustion	Place of Measurement	Debonding Ortho—Tools			ANOVA
		Morita Laser (A) Mean (SD)	Fotona Laser (B) Mean (SD)	LiteTouch Laser (C) Mean (SD)	*p* Values
Saliva ejector	Manikin	29.5 (3.3)	30.5 (1.6) ^a^	31.4 (1.5) ^c^	0.374
	Operator	25.8 (8.5)	30.7 (1.2) ^b^	30.5 (3.0)	A vs. BC *p* < 0.001
	Assistant	26.8 (1.0)	29.4 (1.1)	30.6 (1.9)	A vs. BC *p* < 0.001
High volume evacuator	Manikin	28.4 (3.8)	28.6 (1.3) ^a^	29.6 (2.0) ^c^	0.365
	Operator	26.2 (3.2)	29.2 (0.8) ^b^	30.4 (0.5)	A vs. BC *p* < 0.001
	Assistant	26.5 (1.3)	29.0 (1.1)	29.6 (1.7)	A vs. BC *p* < 0.001
—	—	*p * > 0.05	*p * < 0.05	*p * < 0.05	—

**Table 5 materials-14-02857-t005:** The level of aerosol particles (×103) measured at manikin, operator, and assistant mouths, (Mean (SD)) during endodontic irrigation.

Exhaustion	Place of Measurement	Endo Irrigation—Tools				ANOVA *p*
		Morita Laser (A) Mean (SD)	Fotona Laser (B) Mean (SD)	LiteTouch Laser (C) Mean (SD)	Endodontic Needle (D) Mean (SD)	*p* Values
Saliva ejector	Manikin	30.1 (1.3)	30.3 (1.8)	30.8 (1.6)	31.2 (1.5)	*p* > 0.05
	Operator	30.7 (1.8)	32.0 (1.9)	31.7 (2.1)	31.8 (1.6)	*p* > 0.05
	Assistant	30.3 (1.5)	30.8 (1.7)	31.3 (1.9)	30.7 (1.6)	*p* > 0.05
High volume evacuator	Manikin	29.9 (1.4)	30.8 (0.9)	30.9 (1.7)	30.6 (1.1)	*p* > 0.05
	Operator	30.7 (2.2)	30.7 (2.4)	31.7 (2.1)	31.7 (2.8)	*p* > 0.05
	Assistant	30.1 (1.7)	31.9 (1.4)	30.7 (1.7)	30.9 (1.1)	*p* > 0.05

## Data Availability

Data is contained within the article.

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
