# Peer review of "The Effect of Er:YAG Lasers on the Reduction of Aerosol Formation for Dental Workers"

_materials, 2021, doi:10.3390/ma14112857_

Round 1

Reviewer 1 Report

The authors have prepared a study evaluating the effects of three Er:YAG lasers on the reduction of aerosols in dental cabinets. The paper is innovative and is developed with a proper experimental layout.

However, there are some improvements which can be brought to the current state of the paper:

  • a comparison of the three lasers and reasoning behind choosing them would be appreciated by readers, beyond the information provided in the end of discussions. maybe the information could be summarized in a Table or Figure, if appropriate, and placed in the introduction, materials&methods, or in beginning of the discussions chapter
  • the potential disadvantages of using erbium lasers versus conventional rotary instruments should be briefly presented; also, the risks and counterindications should be mentioned
  • the limitations of the study should be presented succinctly

Author Response

Reviewer: 1

The authors would like to thank the reviewer for their thorough review and recommendations to improve the quality of the manuscript. We have carefully considered your suggestions and implemented the changes into a revised manuscript. In blue are the replies to the reviewers' comments outlining the changes made in the manuscript to address the questions/criticisms of the reviewers.

Reviewer Comments to Author:
The authors have prepared a study evaluating the effects of three Er:YAG lasers on the reduction of aerosols in dental cabinets. The paper is innovative and is developed with a proper experimental layout. However, there are some improvements which can be brought to the current state of the paper: a comparison of the three lasers and reasoning behind choosing them would be appreciated by readers, beyond the information provided in the end of discussions. maybe the information could be summarized in a Table or Figure, if appropriate, and placed in the introduction, materials&methods, or in beginning of the discussions chapter the potential disadvantages of using erbium lasers versus conventional rotary instruments should be briefly presented; also, the risks and counterindications should be mentioned the limitations of the study should be presented succinctly

Add. An additional table describing the features and differences of all three applied devices were described in a new Table1 (M&M section). The study limitations were added at the end of the discussion.

The authors thank the reviewer for his/her comments with great appreciation for the feedback.

Reviewer 2 Report

The topic is very interesting due to the Covid-19 pandemic and will be of interest for Journal readers'. Study design is appropriate, however, Authors should clarify why values of aerosol caused by traditional debonding procedure were not reported within Results section/table 3.

Author Response

Reviewer: 2

The authors would like to thank the reviewer for their thorough review and recommendations to improve the quality of the manuscript. We have carefully considered your suggestions and implemented the changes into a revised manuscript. In blue are the replies to the reviewers' comments outlining the changes made in the manuscript to address the questions/criticisms of the reviewers.

Comments

The topic is very interesting due to the Covid-19 pandemic and will be of interest for Journal readers'. Study design is appropriate, however, Authors should clarify why values of aerosol caused by traditional debonding procedure were not reported within Results section/table 3.

Add. We decided to compare the differences in aerosol generation among the lasers because we assumed that conventional cutting increases the aerosols level similar to caries treatment. However, we must agree with the reviewer that ceramic crown cutting with a bur could show the readers the differences between laser and conventional debonding. Currently, we organizing a similar study in Vivo and we will add also a conventional bur group to our experience. Thank you very much for your valuable remark.

Reviewer 3 Report

Introduction. This investigation is a paper that presents information for researchers in the field of prevention the risk of viral airborne infection in dental workers. A common feature of dental professional activity is working in the environment where human bioaerosols mixed with water sprays, that particles then increase their velocity and scatter in the dental office. Most dental procedures using handpieces generate aerosols. Healthcare professionals have responsibility to introduce updated innovations with safety protocols for prevention of transmission of all infections to patients and staff, and the use of lasers can apply in many dental treatment procedures.

This section is correct.

Materials and methods. The experiment was conducted using a dental manikin head. The protocol includes a aerosol measure. The lasers applied in the study were equipped with a cooling delivery system. All procedures were repeated six times. The quantity of dental aerosol particles was measured during the following dental procedures, caries removal, crown laser and orthodontic brackets debonding, endodontic treatment. During these dental procedures suction systems was used to remove the aerosols produced.

The authors must indicate if this experimental protocol is original ori t is based in others experimental studies because there are not references in this section.

Results. The aerosol levels measured at the manikin, assistant, and operator's mouth were significantly lower for tested laser systems when compared with conventional contra-angle  handpieces during caries removals. crown laser and orthodontic brackets debonding, but not in endodntic treatment.

This section is correct.

Discussion. The results of this study are analised according to several scientific studies. This section is correct.

Conclusions. This section is very long, and repeat aspects of results.

Author Response

Reviewer: 3

The authors would like to thank the reviewer for their thorough review and recommendations to improve the quality of the manuscript. We have carefully considered your suggestions and implemented the changes into a revised manuscript. In blue are the replies to the reviewers' comments outlining the changes made in the manuscript to address the questions/criticisms of the reviewers.

Comments

Introduction. This investigation is a paper that presents information for researchers in the field of prevention the risk of viral airborne infection in dental workers. A common feature of dental professional activity is working in the environment where human bioaerosols mixed with water sprays, that particles then increase their velocity and scatter in the dental office. Most dental procedures using handpieces generate aerosols. Healthcare professionals have responsibility to introduce updated innovations with safety protocols for prevention of transmission of all infections to patients and staff, and the use of lasers can apply in many dental treatment procedures.

This section is correct.

Add. Thank you for the comment.

Materials and methods. The experiment was conducted using a dental manikin head. The protocol includes a aerosol measure. The lasers applied in the study were equipped with a cooling delivery system. All procedures were repeated six times. The quantity of dental aerosol particles was measured during the following dental procedures, caries removal, crown laser and orthodontic brackets debonding, endodontic treatment. During these dental procedures suction systems was used to remove the aerosols produced.

The authors must indicate if this experimental protocol is original or it is based in others experimental studies because there are not references in this section.

Add. The protocol of the study was prepared by the authors of the paper. We added this statement in the M&M section.

Results. The aerosol levels measured at the manikin, assistant, and operator's mouth were significantly lower for tested laser systems when compared with conventional contra-angle  handpieces during caries removals. crown laser and orthodontic brackets debonding, but not in endodntic treatment.

This section is correct.

Add. Thank you for the comment.

Discussion. The results of this study are analised according to several scientific studies. This section is correct.

Add. Thank you for the comment.

Conclusions. This section is very long, and repeat aspects of results.

Add. The conclusion has been shortened

The authors thank the reviewer for his/her comments with great appreciation for the feedback.
